# The impact of withdrawing aquaculture facilities on metazooplankton communities in the lakes are connected to the Yangtze River, China

**Yixing Zhang, Yutao Wang, Zhongze Zhou** *

School of Resources and Environmental Engineering, Anhui University, Hefei, PR China

* azhzz@ahu.edu.cn

## Abstract

The withdrawal of aquaculture facilities has an important impact on the aquatic ecosystem of the lakes connected to the Yangtze River. In order to elucidate the response mechanism of metazooplankton to the changes in water environment after the removal of aquaculture facilities, we collected metazooplankton samples and investigated the water environment in the Huayanghe Lakes from the summer of 2018 to the spring of 2019. Aquatic plants recovered quickly, and water eutrophication was relieved, especially in Lake Huangda, followed by Lake Bo. During our study, the highest regional (γ) diversity was 71 in summer, while the lowest was 32 in winter. Species turnover in space (β diversity) varied between 10.01 and 56.52, which was highest in summer. Based on redundancy analysis, environmental factors such as transparency, Chl α, water temperature and water depth, had greatly effects on the metazooplankton community structure. The results showed that the restoration of aquatic plants increased species diversity and metazooplankton density. This study provides a data basis for lakes restoration and a scientific basis for the management and protection of lakes water ecosystem.

## 1. Introduction

The Anhui section of the Yangtze River basin is located in the mid-subtropical zone, and there are large areas of shallow lakes on both sides. These large shallow lakes along the Yangtze River in Anhui province are important places for fish migration and breeding along the east coast of China, habitats for winter migratory birds, and important gathering places for migratory birds from East Asia to Australia [1]. Since 2000, due to the rapid development of aquaculture industry, the aquatic plants of the lake have been seriously damaged by enclosure aquaculture, especially the submerged plants which have almost disappeared, and the habitat of waterbirds has been seriously degraded, seriously affecting the safety of the entire lake wetland ecosystem [2].

Metazooplankton, mainly composed of rotifera, cladocera and copepoda, feed mainly on smaller phytoplankton, protozoa, bacteria and some organic detritus, and are important

**Data Availability Statement:** All relevant data are within the manuscript and its Supporting Information files.

**Funding:** This work was supported by the Joint Research Project for the Yangtze River Conservation (Phase I), China (No.2019-LHYJ-01-0212, 2019-LHYJ-01-0212-17).

**Competing interests:** The authors have declared that no competing interests exist.

primary consumers in aquatic ecosystems [3, 4]. Metazooplankton is also an economic aquatic animal, which can be used as important food for fish and other large aquatic organisms [5]. Therefore, metazooplankton play an important role in energy flow and material circulation in aquatic ecosystems [6]. And the zooplankton have long been noted as a secondary producer by occupying almost middle positions of the food chain [7]. Compared with other aquatic animals, they are small but numerous, sensitive to the living environment and have strong metabolic activity [8]. Any change in their community structure can effectively reflect the nutrient status of the water body [9, 10]. Changes in metazooplankton community structure may be influenced by many environmental factors in the water environment, including abiotic conditions such as water temperature, water level, light, dissolved oxygen content, aquatic vascular plants in the same habitat, emergent, floating, floating-leaved, and submerged plants, as well as other biological conditions such as phytoplankton, higher aquatic animal (fish, shrimp, etc.) and benthic organisms [11–15].

Huayanghe Lakes in Anhui play an important ecological function in the Yangtze River basin. However, from 2007 to 2017, the rapid development of a fishery gradually destroyed the submerged plants. Since 2018, with the strong support of the national great protection policy for the Yangtze River, the local government removed the aquaculture seine in the Huayanghe Lakes. This resulted in the quick restoration of aquatic plants and an increase in the number of bird populations. Fishes that were cultivated by high-density economic fish in the form of artificial seine converted to natural reproduction, greatly reducing the density. Field survey results find that the rapid restoration of aquatic plants in Lake Huangda in summer is dominated by the floating-leaved plants *Trapa incisa*, whose coverage can reach more than 95%, and the distribution area of aquatic plants such as *Zizania caduciflora* and *Nelumbo nucifera* by the lake can reach 90%. According to the data measured by Zhang et al. [6] secchi depth (SD), total nitrogen (TN), and dissolved oxygen (DO) content of Lake Huangda were 0.37 m, 1.46 mg/L, and 9.40 mg/L respectively; in the 2018 survey, they were 0.48 m, 0.73 mg/L, and 10.00 mg/ L, indicating that the water quality improved after the removal of the seine. Susong County's water production dropped sharply from 90400 t in 2017 to 83900 t in 2018. Based upon the data cited above, further studies should be conducted to clarify the impact of the removal of the fishery purse seine on the metazooplankton in the lakes connected to the Yangtze River.

From the summer of 2018 to the spring of 2019, we monitored the restoration of aquatic plants, parameters of lake environmental factors, composition, density and species diversity of metazooplankton. The aims of this study are as follows: (1) analyze the impact of aquatic plants recovering rapidly after the removal of enclosure aquaculture on water environment and metazooplankton and (2) evaluate the main ecological factors influencing metazooplankton. The study is expected to provide a scientific basis for the restoration and management of lake water ecosystems in the Yangtze River basin.

## 2. Material and method

### 2.1 Description of study area

Huayanghe Lakes (116°00"E-116°33"E, 29°52"N-30°58"N) lie on the north bank of the Yangtze River in Anhui Province, China. The lakes have a surface area of 580 ± 115.8 km$^2$. The average water level is 14.2 m with the average water depth of 4.0 m. Huayanghe Lakes is a natural lake, and its plants distribution pattern was gradually distributed from emergent plants and floating-leaved plants at the lake edge to submerged plants in the lake area. From 2000 to the end of 2017, enclosure aquaculture seriously degraded the aquatic plants of the Huayanghe Lakes.

According to the requirements of the biological water environment monitoring part of "The Environmental Monitoring Technical Specifications" [16] and based on the size and

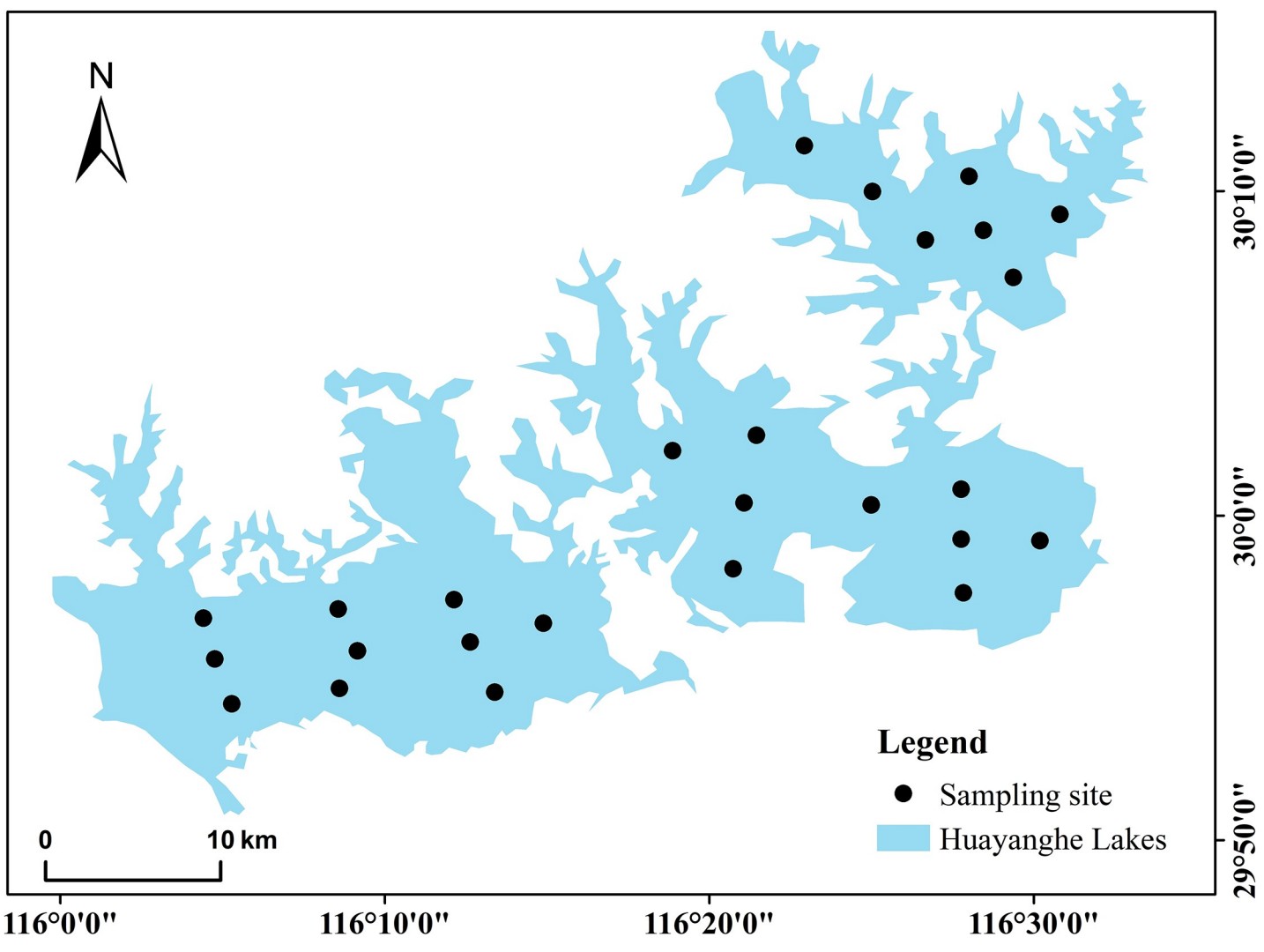

**Fig 1. Distribution map of sampling sites in the Huayanghe Lakes.**

shape characteristics of each lake in the Huayanghe Lakes, we set up a total of 26 sample points (Fig 1, including seven sites in Lake Bo, nine sites in Lake Huangda and ten sites in Lake Long-gan) in the lakes and collected samples once per season from August 2018 to April 2019 (August 2018 is summer, November 2018 is autumn, January 2019 is winter, and April 2019 is spring).

## 2.2 Sampling collection and treatment

We collected the qualitative samples of rotifera by the net towing method (a plankton net with a pore size of 64 μm), and 1L of mixed water samples for the quantitative samples. On site, we added Lugol iodine solution for fixation. We brought the samples back to the laboratory for standing precipitation for 48 h, concentrated to 30 mL and counted them under a microscope.

The collection method of qualitative samples of cladocera and copepoda was the same as that of rotifera qualitative samples, but the water samples were filtered using a 112 μm mesh net and fixed with 4% formaldehyde solution. We collected the quantitative samples by

filtering at least 10 L of mixed water through a 64 μm mesh net, and then concentrated to 50 mL plastic bottles. We immediately added a 4% formaldehyde solution. Using a microscope in the laboratory, we identified and counted the species. The counting method of metazooplankton mainly refers to "Freshwater Plankton Research Method" [17] for counting. The species identification mainly refers to the description of metazooplankton by Wang [18], Koste [19], Jiang and Du [20], and Shen [21].

At each sampling site, we collected an appropriate amount of water samples while taking metazooplankton samples. We stored the water samples at low temperature and brought them back to the laboratory according to the description of SEPB [22] for the measure of total nitrogen (TN), total phosphorus (TP), ammonium nitrogen (AN), nitrate nitrogen (NN) and other indicators. Using the 90% acetone extract+spectrophotometry measurement, we analyzed the Chlorophyll.α (Chl α) concentration, and then measured the turbidity (Turb; Hach, America) with the turbidimeter. We conducted an on-site measurement of water depth (WD), water temperature (WT), transparency (SD; Secchi disk), dissolve oxygen (DO), conductivity (Cond) and other water quality parameters.

## 2.3 Data analyses

In order to describe and explain the difference of the metazooplankton communities among each seasonal sample, we compute the biodiversity indices according to Dube et al. [23]. Another name for γ diversity is regional diversity. This measures the total number of species encountered at each sampling occasion [24]. The second diversity index is α diversity (α). This determines the average number of species under each local habitat [25, 26], also known as local diversity. The last index is β diversity, which represents the species turnover. The value is calculated according to the modified formula of Harrison et al. [27]. The formula is: $\beta = \{[(\gamma/\alpha_{ave})-1]/(N-1)\}^*100$, where $N$ is the number of lakes in each sampling period and $\alpha_{ave}$ is the mean α diversity. It varies between 0 (complete similarity: all regional species occur in all habitats) and 100 (complete dissimilarity: each species occurring in a single habitat).

We computed he dominant species of metazooplankton, determined based on the dominance value of each species, by the formula $Y = (n_i/N)^*f_i$, where $n_i$ is the abundance of individual species $i$, $N$ is the total number of all species, and $f_i$ is the occurrence frequency of species $i$. When $Y \geq 0.02$, this is the dominant species [28].

We plotted the diagrams using Microsoft Excel and Origin and performed a Pearson correlation analysis by applying the SPSS package (Version 20.0, IBM http://www.ibm.com) to evaluate the influence of the limnological variables on metazooplankton densities. To analyze the relationship between the metazooplankton community and the environmental factors, we used CANOCO software. With the ArcGIS software (Version 10.0, ESRI http://www.esri.com), we drew the vegetation distribution map and constructed the interpolation map by the kriging method.

## 3. Results

### 3.1 Ecological factors parameters

Table 1 shows our examination of the physicochemical parameters of the Huayanghe Lakes. There were obvious differences in water temperature in the four seasons, the lowest in winter and the highest in summer, with a range of 4.5~34˚C. The water was slightly alkaline on the whole with a pH value of 8.50±0.43, and there was no significant difference between seasons (p>0.05). The average conductivity was 138.11~184.87 us/cm. In one year, transparency generally increased appreciably with increasing water depth (p<0.05), and turbidity and transparency showed a significant negative correlation (p<0.01).

**Table 1. Seasonal variation of physical and chemical factors (mean±SD) in the Huayanghe Lakes.**

| Parameters | 2019.04/Spring | 2018.08/Summer | 2018.11/Autumn | 2019.01/Winter |
|---|---|---|---|---|
| WT/˚C | 25.03±1.48 | 30.80±1.23 | 15.10±0.90 | 5.31±0.39 |
| pH | 8.78±0.34 | 8.25±0.42 | 8.57±0.36 | 8.38±0.31 |
| DO/(mg/L) | 9.46±1.12 | 8.68±0.77 | 11.09±0.68 | 13.24±0.33 |
| Cond/(us/cm) | 184.87±27.24 | 138.11±24.15 | 156.71±21.38 | 162.46±24.53 |
| SD/cm | 19.65±17.50 | 59.28±40.66 | 57.30±22.11 | 27.54±22.43 |
| DW/m | 1.45±0.41 | 2.98±0.48 | 2.30±0.54 | 2.15±0.49 |
| Turb/NTU | 250.17±262.04 | 28.35±21.56 | 25.19±12.39 | 197.32±121.49 |
| Chl α/(ug/L) | 3.26±2.52 | 5.09±5.19 | 4.28±4.12 | 2.18±1.69 |
| AN/(mg/L) | 0.48±0.41 | 0.08±0.04 | 0.47±0.16 | 0.20±0.10 |
| NN/(mg/L) | 0.18±0.19 | 0.04±0.03 | 0.05±0.02 | 0.13±0.05 |
| TN/(mg/L) | 0.94±0.24 | 0.56±0.32 | 1.06±0.39 | 0.73±0.22 |
| TP/(mg/L) | 0.18±0.10 | 0.01±0.01 | 0.10±0.05 | 0.12±0.04 |

Table 1 shows that the monitoring results of the TP and TN concentrations in each season were considerably different (p<0.05). The concentrations of TN and TP in Lake Huangda were the lowest in summer, at 0.48 mg/L and 0.005 mg/L respectively; the maximum Chl α concentration in Lake Longgan was 9.91 ug/L. The ratio of TN/TP (19.48±18.99) experienced a notable change, and the change of nutrition inevitably affected the growth of phytoplankton, thus affecting the zooplankton community structure. The concentration of Chl α was highest in summer (5.09 ug/L) and lowest in winter (2.18 ug/L).

In 2015 and 2016, the aquatic plants monitoring results for two years recorded that, in summer: Ass. *Nelumbo nucifera* and Ass. *Zizania caduciflora* were a zonal distribution in the coastal waters of Lake Bo; Ass. *Trapa incisa* was mainly distributed in the north and south of Lake Huangda, in the east of Lake Longgan, and the north of Lake Bo. Ass. *Potamogeton wrightii* was distributed sporadically in the middle of Lake Huangda, and the aquatic plants distribution area was not more than 20% (Fig 2A).

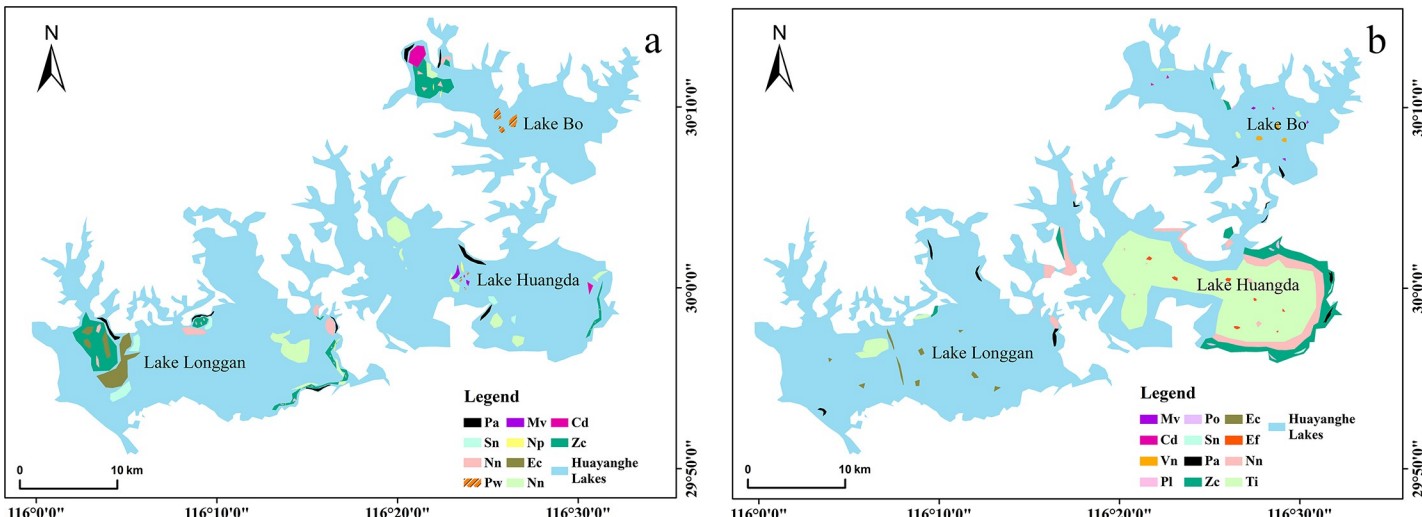

**Fig 2.** Distribution map of main aquatic vegetation in the Huayanghe Lakes was in the summer of 2015(a) and 2018(b). Abbreviations was used in the diagram: Mv: *Myriophyllum verticillatum*; Cd: *Ceratophyllum demersum*; Vn: *Vallisneria natans*; Pl: *Polygonum lapathifolium*; Po: *Polygonum orientale*; Sn: https://p1.ssl.qhimg.com/t0131777b6b3f42e087.jpg*Salvinia natans*; Pa: *Phragmites australis*; Zc: *Zizania caduciflora*; Ec: *Eichhornia crassipes*; Ef: *Euryale ferox*; Nn: *Nelumbo nucifera*; Ti: *Trapa incisa*; Pw: *Potamogeton wrightii*; Np: *Nymphoides peltatum*.

At the end of 2017, the government removed the aquaculture facilities, and the aquatic plants quickly recovered. During the investigation in the summer of 2018, we found that the coverage of aquatic plants in each lake increased significantly. The dominant species in the coastal waters were emergent plants such as *Nelumbo nucifera* and *Zizania caduciflora*, and the accompanying species were floating plants such as *Hydrocharis dubia* and *Salvinia natans*. However, the distribution of the aquatic plants in each lake area was appreciably different. The submerged plants we surveyed near Lake Bo were mainly *Vallisneria natans*, *Ceratophyllum demersum* and *Myriophyllum verticillatum*. There were small areas of *Trapa incisa* and floating *Eichhornia crassipes* in Lake Longgan in the boundary waters of Hubei. In Lake Huangda, however, the floating-leaved plants *Trapa incisa* and *Trapa bispinosa* germinated rapidly and became dominant species, and their distribution area covered 20970 hm$^2$ in Lake Huangda, accounting for more than 87% of the total area of the lake (Fig 2B).

## 3.2 Metazooplankton diversity indices and dominant species

We recorded the highest γ diversity in the summer, with a total of 71 zooplankton taxa: 45 Rotifera, 17 Cladocera, and 9 Copepoda (Fig 3). The lowest γ diversity was 32, including 18 Rotifera, 7 Cladocera, and 7 Copepoda in winter. In addition, the γ diversity in spring (51 taxa: 23 Rotifera, 18 Cladocera, and 10 Copepoda) was higher compared to that in autumn (45 taxa: 25 Rotifera, 12 Cladocera, and 8 Copepoda). Similarly, there was a significant trend for the highest α diversity (52.67±7.53) to be in the summer than during the other sampling seasons. The lowest α diversity was 18.00±3.95 in winter. Species turnover (β diversity) in the space trended first upward, from spring (26.51) to summer (56.51), and then downward, from autumn (14.51) to winter (10.01).

The development of the metazooplankton community and the dominant species groups showed hard variation in each studied season. But, Nauplii was always present in the three lakes on all sampling occasions. And most of the dominant rotifera were the indicator species of the middle fouling zone, such as *Branchionus budapestiensis*, *Keratella cochlearis*, *Brachionus forficula*, etc. We recorded the same dominant groups, including *Trichocerca pusilla*, *Keratella cochlearis*, *Polyarthra trigla* and *Bosmina longirostris*, in the three lakes during the sampling period of the summer of 2018. *Monostyla lunaris*, found only in Lake Longgan, contributed the most among the dominant species. In the spring of 2019, we documented more

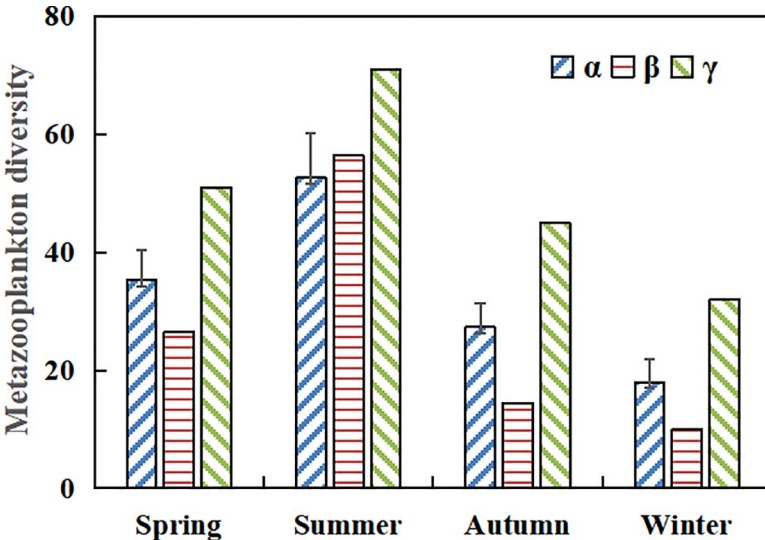

**Fig 3. The diversity index of metazooplankton taxa in the Huayanghe Lakes collected in different seasons.**

*Branchionus budapestiensis* and *Ascomorpha ecaudis* than other species with *Daphnia cucullata* also was recorded in samples. Throughout the lakes surveyed in the autumn of 2018, we observed *Keratella quadrata*, *Ascomorpha saltans*, *Ascomorpha ecaudis*, and *Microcyclops vaticans*. In addition, *Keratella quadrata*, *Polyarthra trigla*, *Brachionus angularis*, *Branchionus clycifolrus* and *Asplachna priodonta* dominated the metazooplankton groups in the studied lakes in the winter of 2018.

### 3.3 Temporal and spatial variation in metazooplankton community

Rotifera always dominated the metazooplankton community during the sampling phases. The density accounted for more than 80% at each season (Fig 4). The relative density of cladocera was highest in the spring (2.65%), while that of copepoda was highest in the autumn (11.91%). We found the higher mean cladocera density (56.67 ind./L and 25.36 ind./L) in the spring and summer, and watched it gradually decrease to its lowest in winter. The mean copepoda density ranged from 12.94 ind./L in the spring to 24.11 ind./L in the autumn. The total rotifera density was significantly higher than that of the other two crustacean zooplankton (p<0.01), within the range of 170.54–2723.20 ind./L.

As shown in Fig 5, there was a significant spatial difference in the annual average density of metazooplankton in different lakes. The level of metazooplankton in Lake Longgan was generally on the high value and gradually declined along a certain gradient; the highest annual average density was 4361.0 ind./L. While the overall level of Lake Bo was low, the lowest value was 294.6 ind./L.

### 3.4 Relationship between metazooplankton and environmental factors

Using Canoco, we first performed a detrended correspondence analysis (DCA) between the densities of 17 major metazooplankton species and environmental factors. The results showed that the length of the maximum gradient in the sequencing axis was 2.1 (<3), which means that there was a linear relationship between metazooplankton and environmental factors. So, we applied redundancy analysis (RDA). After the Monte Carlo replacement detection, we performed an RDA between main species and environmental factors.

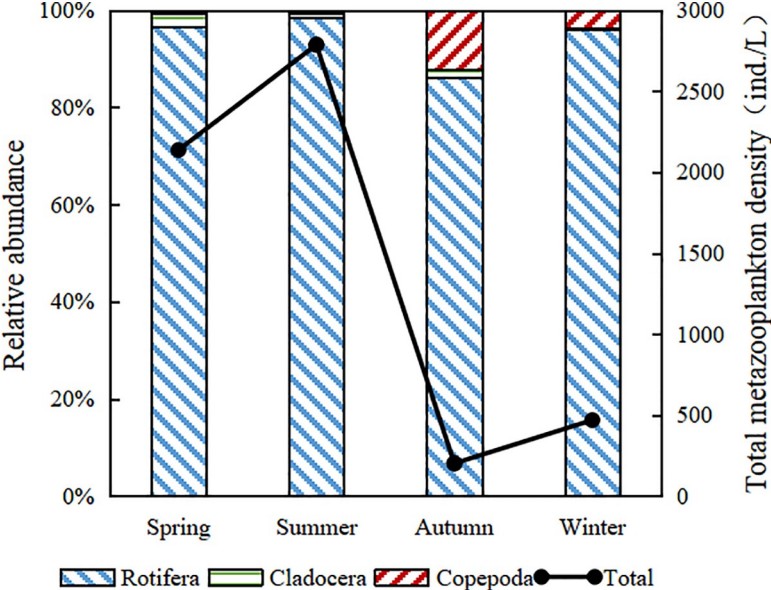

**Fig 4. Temporal variation of metazooplankton density in the Huayanghe Lakes.**

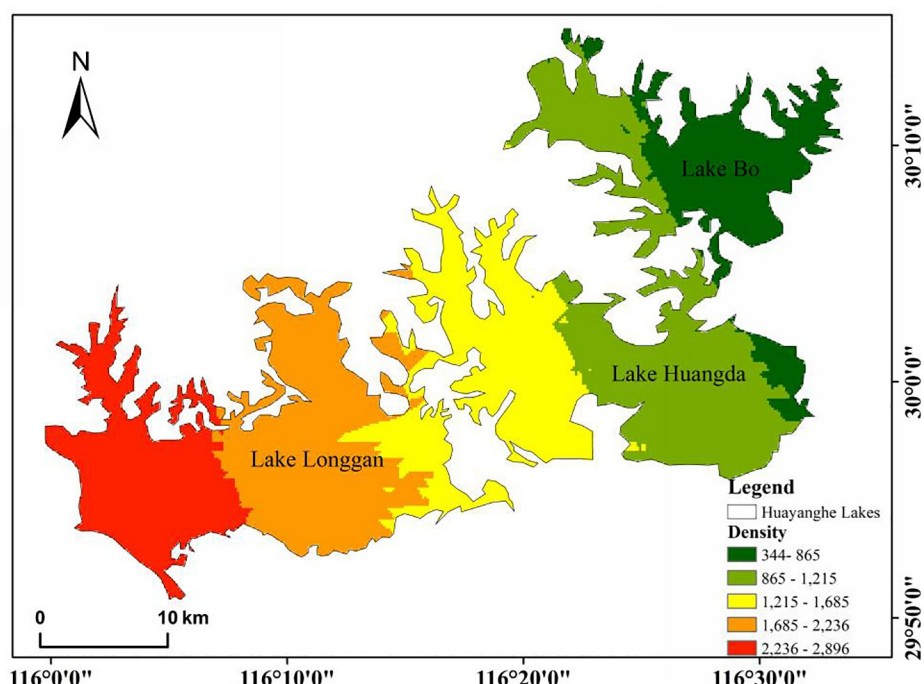

**Fig 5. Distribution of metazooplankton density at three studied lakes.** (The plot is expressed in terms of the annual average metazooplankton density, the unit is ind./L).

RDA results exhibited that the eigenvalues of axis 1 and axis 2 were 0.4152 and 0.0836, respectively. The linear combinations of environmental factors making up the first two RDA axes explained the 31.9% of the metazooplankton assemblage data variance. It can be seen from Fig 6 that transparency (38.6%), water depth (12.9%), and Chl α (11.1%) are closely related to the community structure and annual average density of metazooplankton. Chl α, TN, and Cond positively affected *Monostyla lunaris*, *Conochilus hippocrepis*, and the mean annual density of metazooplankton, but negatively impacted *Thermocyclops hyalinus*, which may make it difficult to sustain the mass reproduction of phytoplankton. WT was the main influencing factor for *Keratella cochlearis* and *Keratella quadrata*. TP and pH had a positive influence on *Ascomorpha saltans*, *Ascomorpha ecaudis*, *Microcyclops vaticans* and *Cyclops vicinus*, but these four species struggled with SD and might prefer to live in water with a high degree of nutrition. Therefore, TN, TP and Cond were also important factors affecting the metazooplankton community in the Huayanghe Lakes.

## 4. Discussion

### 4.1 Effect of the removal of enclosure aquaculture on aquatic plants and water environment

Our investigation of the Huayanghe Lakes after the removal of enclosure aquaculture facilities revealed the aquatic plants restoration of each sub-lake was obviously different. Submerged plants were the main species in Lake Bo, and floating-leaved plants were the main species in Lake Huangda, while there were fewer aquatic plants in Lake Longgan. This was different from before enclosure aquaculture (Fig 2). The plants mainly grew along the lakeshore and river mouth, and the submerged plants basically disappeared in the open water area of each lake.

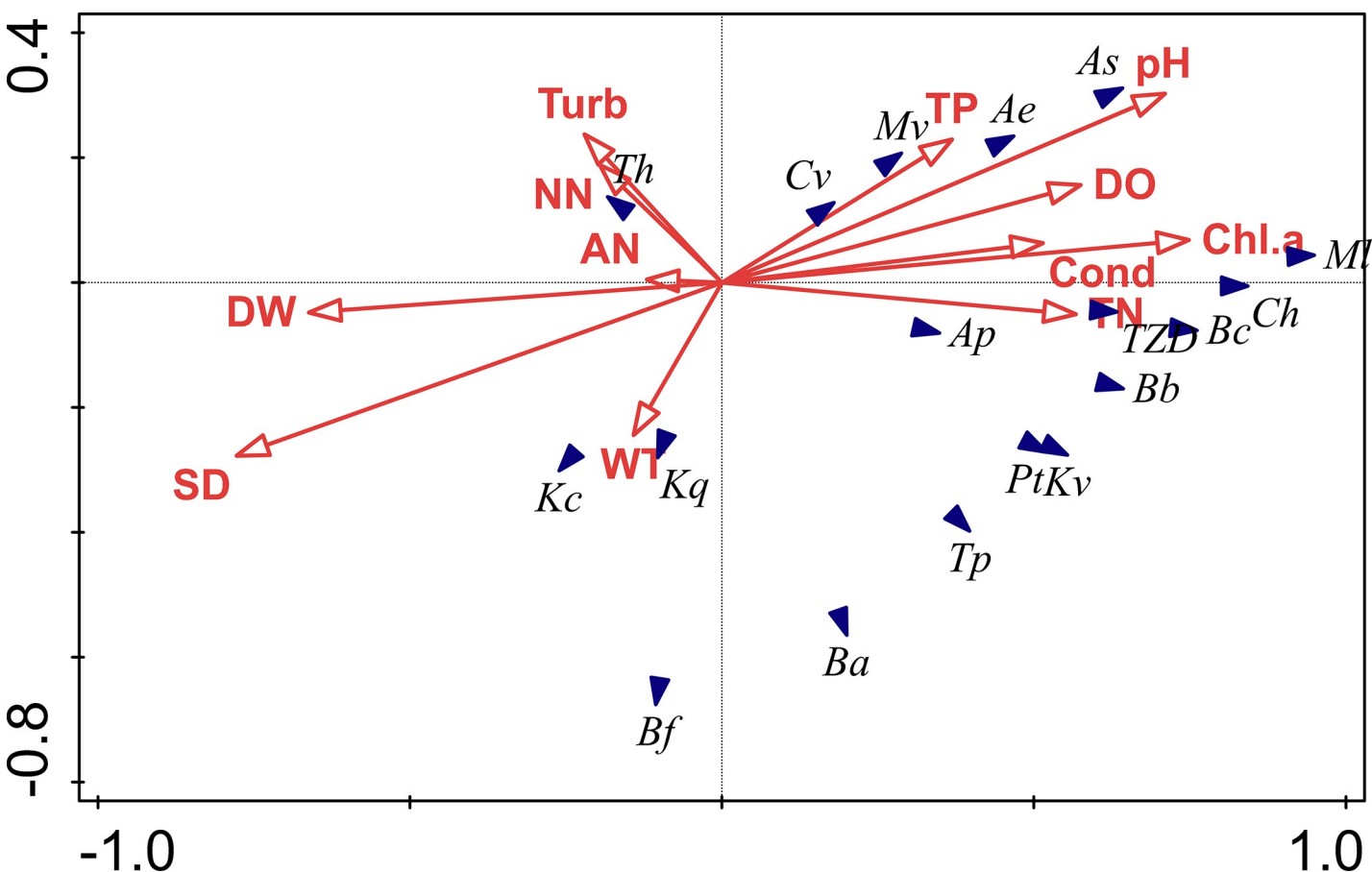

**Fig 6. Biplot diagram for redundancy analysis between major metazooplankton species (solid arrowhead) and environment factors (hollow arrow with line) in the Huayanghe Lakes.** Abbreviations was used in the diagram: *Kc*: Keratella cochlearis, *Pt*: Polyarthra trigla, *Kq*: Keratella quadrata, *Bc*: Branchionus clycifolrus, *Tp*: Trichocerca pusilla, *Ml*: Monostyla lunaris, *Kv*: Keratella valga, *Ba*: Brachionus angularis, *Bf*: Brachionus forficulav, *Bb*: Branchionus budapestiensis, *Ch*: Conochilus hippocrepi, *Ae*: Ascomorpha ecaudis, *As*: Ascomorpha saltans, *Ap*: Asplachna priodonta Gosse, *Th*: Thermocyclops hyalinus, *Mv*: Microcyclops vaticans, *Cv*: Cyclops vicinus, TZD: Annual mean density.

The reason for this might be that there was a reduction of fish density along the shoreline and river mouth, especially the grass-feeding economic fish, which was mainly composed of *Ctenopharyngodon idella*, *Parabramis pekinensis*, and *Megalobrama amblycephala*. With fewer of these fish to feed on the fruits and seedlings of aquatic plants, the possibility of their germination and growth increased [29]. High-density net-enclosed fish culture affected the surrounding environment and aquatic plants due to its pollution effect and physical barrier effect, causing vegetation degradation [30]. On the other hand, habitat connectivity can maintain the integrity of wetland ecosystem and encourage the diversity of aquatic organisms. Aquatic plants transform water and inorganic nutrients (mainly nitrogen and phosphorus) into organic matter through photosynthesis, while becoming food for consumers themselves. Aquatic plants and phytoplankton together constitute the primary producer in the lake ecosystem, which are the starting point of the food chain and play an important role in the material cycle and energy flow. Therefore, aquatic plants have an important influence on the structure and function of the lake ecosystem [31, 32].

Compared with previous results [6, 33], the results of this study showed that the contents of TN and TP decreased significantly in summer when aquatic plants grew vigorously, but showed an upward trend throughout the year. Xue et al. [34] studied the Lake Dianshan after

the demolition of the aquaculture facilities, and also found that the concentration of TN and TP in the water increased overall. It might be due to the presence of large aquatic plants, which absorbs and assimilates nutrients such as carbon, nitrogen and phosphorus during its growth [35], fixed sediments and mud, and reduces wind and wave action [36]. However, in the process of natural succession and seasonal changes, the growth decline, death and litter decomposition of large aquatic plants are necessary stages; and the decomposition of their residues and the release of nutrients lead to the increase of nutrients [37, 38]. Kosten et al. [39] demonstrated that the presence of aquatic plants can cause significant small-scale reductions in the nutrients, Chl $\alpha$ and Trub in or above the plants layer compared to sites that don't have plants.

Chl $\alpha$ represents phytoplankton biomass, and the highest Chl $\alpha$ was detected in Lake Longgan. The field investigation observed that there was almost no large submerged plants in Lake Longgan, which was a mainly shallow lake dominated by phytoplankton with turbidity state, and relatively poor water quality. As phytoplankton covered almost the whole lake, light could not enter the water, and aquatic plants seedlings were unable to photosynthesize. This severely inhibited the growth of aquatic plants. Huo et al. [40] carried out ecological restoration in Lake Dishui, Shanghai, using *Daphnia magna* to control algal propagation and increase water transparency, and successfully cultivated large aquatic plants. We observed small areas of *Trapa incisa* and scattered floating *Eichhornia crassipes* in the Hubei boundary of Longgan Lake. The study by Wang et al. [41] mentioned that floating plants had low environmental requirements, were pollution-resistant, and could grow in an environment with high nitrogen and phosphorus levels, especially some fast growing floating plants, such as *Eichhornia crassipes* and *Lemna minor*. They are widely used to treat eutrophication in water bodies to reduce nitrogen and phosphorus levels, improve water transparency, and gradually restore other aquatic plants and the whole ecological environment [41, 42].

## 4.2 Effects of aquatic plants restoration on metazooplankton after the removal of enclosure aquaculture

The distribution of metazooplankton communities depends to a large extent on the micro-environmental characteristics of water bodies, and large aquatic plants can provide micro-environments (such as space, food resources, etc.) for metazooplankton [43]. In addition to their ability to rapidly absorb nutrients from water and sediment, aquatic plants can alter some of the physical and chemical environmental factors that impact the distribution of metazoo-plankton, such as transparency, pH, disturbance, etc.; it can influence phytoplankton, which are important food sources for metazooplankton [44]; it also provides shelter for metazoo-plankton [36], thus altering fish feeding [45]. Therefore, aquatic plants can affect the habitats of metazooplankton due to many aspects, and are one of the important factors that impact the species composition, density and diversity of metazooplankton [46]. This study identified a total of 82 species of metazooplankton, including 49 species of rotifera, 33 species of cladocera and copepoda, with an average density of 202.39–2786.70 ind./L. According to the data measured by Xu [33] from April 2015 to January 2016, a total of 53 species of metazooplankton were discovered in the same lake, including 39 species of rotifera, with an average density of 223.08–2061.62 ind./L. During the two years from April 2015 to January 2017, Zhang et al. [6] discovered 25 species of crustacean zooplankton. This indicated that, on the whole, the species diversity and density of metazooplankton were on the rise. The number of large crustaceans also increased significantly, but rotifera were still the dominant species. This may be because that, in the early stage of submerged plants recovery, planktonic algae are still the main primary producers and that the secondary producers depending on phytoplankton still exist in large numbers [47, 48]. According to some researchers [49, 50], the species abundance and

diversity of metazooplankton were generally higher in waters with aquatic plants coverage than in those without plants. Hu et al. [51] showed that in lakes with different aquatic plants coverage, the species diversity and density of metazooplankton would increase with the rise of aquatic plants coverage.

In our study, the average annual density of metazooplankton had a gradient variation (Fig 5) with a high β diversity (>10), indicating that there were significant spatial differences in the metazooplankton community; especially in summer, species richness of crustacean zooplankton were higher in Lake Bo (26 species) and Lake Huangda (21 species) than in Lake Longgan (18 species). Lake Longgan had the lowest vegetation coverage among the Huayanghe Lakes, and the submerged plants had a better vegetation recovery in Lake Bo, while the floating-leaved plants coverage was the highest in Lake Huangda. This is consistent with the results of other scholars [8, 12], and the proportion of cladocera and copepoda in the abundance of metazooplankton in Lake Bo is higher. That is, the species richness of macrozooplankton is significantly enhanced during the restoration of aquatic plants. Compared with Xu [33], *Sida crystallina* and *Bosminopsis deitersi* were widely distributed, and the species number of cladocera rose (Daphnia from the original only *Daphnia cucullata* to the presence of *Daphnia magna*, *Daphnia obtusa*, *Daphnia hyalina* and *Daphnia pulex*). This may be due to the existence of large aquatic plants and the expansion of coverage, coupled with the obvious increase of water transparency in Lake Huangda and Lake Bo. It has been uncovered that large-size species like to grow in aquatic plants, such as *Sida crystallina* and *Bosminopsis deitersi*, which appear in demonstration areas of aquatic plants restoration [52]. Zhang et al. [53] proposed that the change of aquatic plants coverage was one of the important factors affecting the species composition of the cladocera community, and that lush aquatic plants could provide shelters for cladocera. After the removal of enclosure aquaculture, the indicator species of the middle fouling zone, such as *Branchionus budapestiensis*, *Brachionus forficula* and *Keratella cochlearis* still dominated the metazooplankton communities [54]. However, compared with Zhang et al. [6] who recorded that $\gamma_{max}$ was 52 (including Lake Wuchang) in May 2016, the $\gamma_{max}$ diversity (71) in this study has markedly increased.

## 4.3 Effects of fish on metazooplankton after the removal of enclosure aquaculture

The fish is located at the top of the food chain of the aquatic ecosystem. This has an important influence on the structure change of metazooplankton community. The research of Yang et al. [55] proved that fish preyed on metazooplankton selectively and generally gave preference to larger individuals under the same conditions. Susong County's water production was 90400 t in 2017, compared with 83900 t in 2018. After the removal of the purse seine, there was an obvious decrease in the output of fish. This reduced the predation pressure on the metazooplankton. Both the abundance of metazooplankton and the species of cladocera and copepoda increased markedly. We observed some species, such as *Daphnia magna*, *Daphnia hyalina*, *Sida crystallina*, *Cyclops vicinus* and *Microcyclops intermedius*. Fish stocking density also plays an important role in zooplankton community structure [56, 57].

## 4.4 Effect of water environment improvement on metazooplankton after the removal of enclosure aquaculture

Pearson correlation analysis and RDA displayed the fact that the main physical and chemical factors affecting the structure of the metazooplankton community in the Huayanghe Lakes after the removal of purse seine were SD, water temperature, Chl α, electrical conductivity, etc., which was consistent with many other research results [58, 59].

  

The restoration of aquatic plants led directly, or indirectly, to the changes of physicochemical factors in the Huayanghe Lakes, which led to the transformation of the structure of the metazooplankton community. WT was an important environmental factor affecting the growth, development, community composition, and population distribution of metazooplankton [60, 61]. During this study, we found that the species of rotifera impacted the number of metazooplankton species, with the density of rotifera accounting for more than 80% at each season (Fig 4). The variation trend of the two was basically the same, which was the same as the research results of other periods of the Huayanghe Lakes [6, 33]. This research area had a humid subtropical climate with obvious monsoon and low water temperature in autumn and winter. These conditions are not conducive to the growth and reproduction of metazooplankton, and the species and density are relatively few. However, with the increase of water temperature, resting eggs continued to hatch, Nauplius appeared in large numbers, and cladocera and copepoda species increased. Lin et al. [14] uncovered that metazooplankton was unstable during different sampling cycles in Lake Dishui. From winter to summer, with the increase of temperature and the decrease of turbidity, the species of metazooplankton increased. Meanwhile, their density increased as well. From summer (WT = 30.80°C) to autumn (WT = 15.10°C), there was a significant decrease in the density and γ diversity of metazooplankton. This may be due to seasonal changes in summer and autumn and large changes in water temperature. Through RDA and Pearson correlation analysis, we turned up that transparency and water depth had significant effects on the density of the major species. Hart's [62] study demonstrated that metazooplankton was closely related to transparency and turbidity. The low transparency could inhibit the development of metazooplankton [10, 40]. Some researchers [40, 41] had discovered that the increase in metazooplankton grazing can help improve transparency.

In this study, Chl α concentration was also one of the important influencing factors, and RDA diagram showed that most metazooplankton species and the annual average density pronouncedly correlated with Chl α concentration, TN and TP. Phytoplankton is one of the important food sources of zooplankton, and nutrients such as nitrogen and phosphorus affect metazooplankton mainly by affecting phytoplankton [63, 64].

## 5. Conclusion

This study showed that the withdrawing of aquaculture facilities in the Huayanghe Lakes accelerated the restoration and reconstruction of aquatic plants, promoted the improvement of water quality and increased the γ diversity of metazooplankton, especially in Lake Bo with better restoration of aquatic plants. In order to effectively mitigate water eutrophication and improve biodiversity, we recommend that aquaculture facilities be completely dismantled and aquatic plants restored. Studies have shown that the restoration of aquatic plants improves the transparency of water bodies, increases the diversity of metazooplankton species, and increases the density of metazooplankton. However, in the early stage of aquatic plants recovery, the main plants were floating-leaved plants, floating plants and emergent plants, while submerged plants had relatively poor recovery. In order to restore a more stable ecosystem, artificial planting of submerged plants can be combined with management and maintenance to optimize the structure of aquatic vegetation, protect the diversity of aquatic plants and ultimately reduce the adverse effects of lake eutrophication.

## Supporting information

**S1 File.**
(RAR)

## Acknowledgments

We thank Kun Zhang, An Li, and Yanling Guo for their participation in the field work. We would also like to thank Dr. Chunlin Li and Xia Wan for their advice on the paper. We would like to thank Marci Baun from University of California, Los Angeles for editing the paper.

## Author Contributions

**Funding acquisition:** Zhongze Zhou.

**Investigation:** Yutao Wang.

**Methodology:** Zhongze Zhou.

**Writing – original draft:** Yixing Zhang.

**Writing – review & editing:** Yixing Zhang, Zhongze Zhou.

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
