## [Decision Letter · Decision Letter 0]

20 Mar 2021

PONE-D-21-05633

The effect of the removal of enclosure aquaculture on metazooplankton communities in the lakes connected to the Yangtze River in China

PLOS ONE

Dear Dr. Zhou,

Thank you for submitting your manuscript to PLOS ONE. After careful consideration, we feel that it has merit but does not fully meet PLOS ONE’s publication criteria as it currently stands. Therefore, we invite you to submit a revised version of the manuscript that addresses the points raised during the review process.

Please consider all the comments made by the reviewers. Your revised manuscript will undergo a second round of reviewing.

We look forward to receiving your revised manuscript.

Kind regards,

SSS Sarma

Academic Editor

PLOS ONE

Additional Editor Comments:

I have got the two reviews on your contribution. Both reviewers have unanimously agreed that the manuscript requires a major revision. I with them. So, please revise your manuscript based on the constructive comments offered by the reviewer. The revised manuscript will undergo a second round of reviewing, before a decision is made on the fate of your contribution.

Sincerely

SSS Sarma, Handling Editor

Journal Requirements:

2. In your Methods section, please provide additional location information of the study sites, including geographic coordinates for the data set if available.

4. We suggest you thoroughly copyedit your manuscript for language usage, spelling, and grammar. If you do not know anyone who can help you do this, you may wish to consider employing a professional scientific editing service.  

5. Our internal editors have looked over your manuscript and determined that it is within the scope of our Freshwater Ecosystems Call for Papers. This collection of papers is headed by a team of Guest Editors for PLOS ONE (https://collections.plos.org/s/freshwater-ecosystems). The Collection will encompass a diverse range of research articles on freshwater ecology, including lake ecology. Additional information can be found on our announcement page: https://collections.plos.org/s/freshwater-ecosystems.

If you would like your manuscript to be considered for this collection, please let us know in your cover letter and we will ensure that your paper is treated as if you were responding to this call. If you would prefer to remove your manuscript from collection consideration, please specify this in the cover letter.

6. We note that the grant information you provided in the ‘Funding Information’ and ‘Financial Disclosure’ sections do not match.

7. In your Data Availability statement, you have not specified where the minimal data set underlying the results described in your manuscript can be found. PLOS defines a study's minimal data set as the underlying data used to reach the conclusions drawn in the manuscript and any additional data required to replicate the reported study findings in their entirety. All PLOS journals require that the minimal data set be made fully available. For more information about our data policy, please see http://journals.plos.org/plosone/s/data-availability.

8. We note that Figures 1, 2, 5 in your submission contain map images which may be copyrighted. All PLOS content is published under the Creative Commons Attribution License (CC BY 4.0), which means that the manuscript, images, and Supporting Information files will be freely available online, and any third party is permitted to access, download, copy, distribute, and use these materials in any way, even commercially, with proper attribution. For these reasons, we cannot publish previously copyrighted maps or satellite images created using proprietary data, such as Google software (Google Maps, Street View, and Earth). For more information, see our copyright guidelines: http://journals.plos.org/plosone/s/licenses-and-copyright.

8.1.    You may seek permission from the original copyright holder of Figures 1, 2, 5 to publish the content specifically under the CC BY 4.0 license. 

8.2.    If you are unable to obtain permission from the original copyright holder to publish these figures under the CC BY 4.0 license or if the copyright holder’s requirements are incompatible with the CC BY 4.0 license, please either i) remove the figure or ii) supply a replacement figure that complies with the CC BY 4.0 license. Please check copyright information on all replacement figures and update the figure caption with source information. If applicable, please specify in the figure caption text when a figure is similar but not identical to the original image and is therefore for illustrative purposes only.

Reviewers' comments:

Reviewer's Responses to Questions

**Comments to the Author**

1. Is the manuscript technically sound, and do the data support the conclusions?

Reviewer #1: Partly

Reviewer #2: Yes

2. Has the statistical analysis been performed appropriately and rigorously? 

Reviewer #1: Yes

Reviewer #2: Yes

3. Have the authors made all data underlying the findings in their manuscript fully available?

Reviewer #1: Yes

Reviewer #2: Yes

4. Is the manuscript presented in an intelligible fashion and written in standard English?

Reviewer #1: Yes

Reviewer #2: No

5. Review Comments to the Author

Reviewer #1: PONE-D-21-05633

Full Title: The effect of the removal of enclosure aquaculture on metazooplankton communities in the lakes connected to the Yangtze River in China

Suggestions/revision

Introduction

Original: in aquatic ecosystems (Zhang, 2019).

Suggestion: in aquatic ecosystems (Zhang, 2019). And, the zooplankton have long been noted as a secondary producer by occupying almost middle positions of the food chain (Tugyan and Bozkurt, 2019). Compared with other aquatic….

You can add the section as literature in yellow. International literature contribution

Tugyan, C., Bozkurt, A., 2019. A Study on Zooplankton Fauna and Some Water Quality Parameters of Kozan Dam Lake (Adana, Turkey). Journal of Limnology and Freshwater Fisheries Research 5(3): 147-158.

Original: 68 line (http://www.susongbbs.com/1408365-1-1.html)

69 line (http://www.ahmhxc.com/tongjigongbao/14909_2.html).

Suggestion: these links show in Chinese. It doesn't make sense for the reader. Because not everyone knows Chinese. Please remove the link.

Sampling collection and treatment

Original 106 line and Du (1979), and Shen (1979).

Suggestion: Why did you only use Chinese researchers in the reference books you use for species diagnosis?

Please include here other authors from other countries with international books. For example; Koste 1978……..

(Koste, W., 1978. Rotatoria, Die Rädertiere Mitteleuropas Ein Bestimmungswerk, Begründet von Max Voigt Überordnung Monogononta. I Textband. Gebrüder Borntraeger, Berlin, Stuttgart. 672P and II Textband. 234P.)

Results

Original 161 line summer, Ass. N. nucifera and Ass. Z. caduciflora

Suggestion: Ass. Nelumbo nucifera and Ass. Z. caduciflora (write the expansion of the species)

Original 162 line waters of Lake Bo; Ass. T. incisa

Suggestion: waters of Lake Bo; Ass. T. incisa (write the expansion of the species)

Original 163 line Ass. Potamogeton Wrightii

Suggestion: Ass. Potamogeton wrightii

Original 174 line There were small areas of T. incisa

Suggestion: There were small areas of T. incisa

Original 194 line Trichocerca pusilla, K. cochlearis, P. trigla and

Suggestion: Trichocerca pusilla, K. cochlearis, P. trigla and (write the expansion of the species)

Original 200 line In addition, K. quadrata, P. trigla, B. angularis,

Suggestion: In addition, K. quadrata, P. trigla, B. angularis, (write the expansion of the species)

Original 230 line M. lunaris, B. forficula, and

Suggestion: M. lunaris, B. forficula, and (write the expansion of the species)

Original 231 line correlated with Chl α, TN, and Cond, while T. hyalinus

Suggestion: correlated with Chl α, TN, and Cond, while T. hyalinus (write the expansion of the species)

Original 233 line the main influencing factor for K. cochlearis and K. quadrata. A. saltans, A. ecaudis

Suggestion: the main influencing factor for K. cochlearis and K. quadrata. A. saltans, A. ecaudis (write the expansion of the species)

Original 234 line T. hyalinus were positively

Suggestion: T. hyalinus were positively

Original 283 line We observed small areas of T. incisa and 283 scattered floating E. crassipes

Suggestion: We observed small areas of T. incisa and 283 scattered floating E. Crassipes (write the expansion of the species)

Original 331 line Sida erystallina and

Suggestion: Sida crystallina and

Original 332 line (Daphnia from the original only

Suggestion: (Daphnia from the original only

Original 337 line as S. erystallina and

Suggestion: as S. crystallina and

Original 337 line dominated by the indicator species of the middle fouling zone, such as B. budapestiensis, ,

Suggestion: dominated by the indicator species of the middle fouling zone, such as B. budapestiensis, B. forficula (write the expansion of the species)

Original 354 line New species, such as D. magna, D. hyalina, S. erystallina,

Suggestion: New species, such as D. magna, D. hyalina, S. crystallina,

Original 269 line (Meyer et al., 2019).

Suggestion: (Meyer et al., 2018). 2018 written in references. Correct in the text as above.

Citations in the text should be given in order according to the years. Please apply.

Original: 35 line affecting the safety of the entire lake wetland ecosystem (Zhu et al., 2010).

Suggestion: Include this resource in the References section. Not seen in references.

Original: 284 line The study by Jiang et al. (2008) mentioned that

Suggestion: Include this resource in the References section. Not seen in references.

Original: 328 line Zeng Lei et al. (2018) reported in their study

Suggestion: Include this resource in the References section. Not seen in references.

Notes: reference section shortcomings are noted below.

442 line Huang, X.F., 2000. Investigation, observation and analysis

(This article is not cited in the text. Check it in the text)

506 line Wang, H.J., Ding, X.S., Tan, W.J., Zhou Y.L., 2008. Study on the effects of floating plants (This article is not cited in the text. Check it in the text)

515 line Xie, H., Jiang, Z.G., Xia, Z.J., Guo, W.Y., 2018. Functional groups of fish community in (This article is not cited in the text. Check it in the text)

535 Zhu, W.Z., Zhou, L.Z., 2010. Biodiversity and Conservation in Anqing Floodplain (This article is not cited in the text. Check it in the text)

535 Zhu, W.Z., Zhou, L.Z., 2010. Biodiversity and Conservation in Anqing Floodplain (This article is not cited in the text. Check it in the text) Include this resource in the References section. Not seen in references.

Best regards,

Reviewer #2: I found this research manuscript very interesting and I believe it has scientific impacts. The objectives are very clearly defined. Enough dedications and efforts are observed in results and findings. Authors tried a lot to make this research statistically sound and presentable. Therefore, I have some critical comments on it. Firstly, The discussion need to improve a lot. Please consider my comments very positively. If you reform and rewrite the discussion...then this manuscript could be a master piece. Please avoid gerund sentence (starting with verb+ing). Please read more related articles and rewrite your discussion. Please read and cite more related references in discussion. I expected a conclusion but I didn't find it. Please write a nice conclusion. After all of your revision activities, at the last please write the abstract again. Please follow professional English to make your article more convenience to readers. Trust me you will get more positive feedback on your article if you consider these comments.

If possible try to contact a person who help you in professional English. Please don't feel hesitate on it. I found your research article is very helpful to us but need to be more readable and clear. Hopefully, you got my messages.

6. PLOS authors have the option to publish the peer review history of their article (what does this mean?). If published, this will include your full peer review and any attached files.

Reviewer #1: No

Reviewer #2: **Yes: **Najmus Sakib Khan

---

## [Author Response · Author response to Decision Letter 0]

22 Apr 2021

I didn't find that I need to respond to specific reviewer and editor comments.

---

## [Decision Letter · Decision Letter 1]

10 May 2021

The impact of withdrawing aquaculture facilities on metazooplankton communities in the lakes are connected to the Yangtze River, China

PONE-D-21-05633R1

Dear Dr. Zhou,

We’re pleased to inform you that your manuscript has been judged scientifically suitable for publication and will be formally accepted for publication once it meets all outstanding technical requirements.

Kind regards,

SSS Sarma

Academic Editor

PLOS ONE

Additional Editor Comments (optional):

One reviewer suggested a few minor corrections. These can be added at the proof stage.

Reviewers' comments:

Reviewer's Responses to Questions

**Comments to the Author**

1. If the authors have adequately addressed your comments raised in a previous round of review and you feel that this manuscript is now acceptable for publication, you may indicate that here to bypass the “Comments to the Author” section, enter your conflict of interest statement in the “Confidential to Editor” section, and submit your "Accept" recommendation.

Reviewer #1: All comments have been addressed

Reviewer #2: All comments have been addressed

2. Is the manuscript technically sound, and do the data support the conclusions?

Reviewer #1: Yes

Reviewer #2: Yes

3. Has the statistical analysis been performed appropriately and rigorously? 

Reviewer #1: Yes

Reviewer #2: Yes

4. Have the authors made all data underlying the findings in their manuscript fully available?

Reviewer #1: Yes

Reviewer #2: Yes

5. Is the manuscript presented in an intelligible fashion and written in standard English?

Reviewer #1: Yes

Reviewer #2: Yes

6. Review Comments to the Author

Reviewer #1: Thank you to the authors for making the necessary corrections. I hope the study will contribute to the literature and science. Good work

Reviewer #2: Dear authors,

I appreciate your efforts on your manuscript. I tried to make some comments on your manuscript.

a) No need to delete any references, just try to know how to cite it according to PLOS ONE Style.

b) I marked some lines for rewriting and checking the information (Please see the attached file)

c) Please thoroughly check the references in text.

d) I think you need to improve your discussion little more. Try to make it more concrete. You have to focus on your writing actually. It's readable but I expect more comfortability in your text. If possible, try to add some more references to make your discussion strong. Moreover, try to explain your findings more confidently. I hope then your manuscript will be a masterpiece.

7. PLOS authors have the option to publish the peer review history of their article (what does this mean?). If published, this will include your full peer review and any attached files.

Reviewer #1: No

Reviewer #2: **Yes: **Najmus Sakib Khan

---

## [Editor Report · Acceptance letter]

17 May 2021

PONE-D-21-05633R1 

The impact of withdrawing aquaculture facilities on metazooplankton communities in the lakes are connected to the Yangtze River, China 

Dear Dr. Zhou:

I'm pleased to inform you that your manuscript has been deemed suitable for publication in PLOS ONE. Congratulations! Your manuscript is now with our production department. 

Kind regards, 

on behalf of

Professor SSS Sarma 

Academic Editor

PLOS ONE